# Spatiotemporal Consensus with Scene Prior for Unsupervised Domain Adaptive Person Search

**Yimin Jiang**[†]
Dalian Maritime University
yimin_jiang@dlmu.edu.cn

**Huibing Wang**[*†]
Dalian Maritime University
huibing.wang@dlmu.edu.cn

**Jinjia Peng**[†]
Hebei University
pengjinjia@hbu.edu.cn

## Abstract

Person Search aims to locate query persons in gallery scene images, but faces severe performance degradation under domain shifts. Unsupervised domain adaptation transfers knowledge from the labeled source domain to the unlabeled target domain and iteratively rectifies the pseudo-labels. However, the pseudo-labels are inevitably contaminated by the source-biased model, which misleads the training process. This, in turn, reduces the quality of the pseudo-labels themselves and ultimately affects the search performance. In this paper, we propose a Spatiotemporal Consensus with Scene Prior (STCSP) framework that effectively eliminates the interference of noise on pseudo-labels, establishes positive feedback, and thus gradually bridging the domain gap. Firstly, STCSP uses a Spatiotemporal Consensus pipeline to suppress the noise from being mixed into the pseudo-labels. Secondly, leveraging the scene prior, STCSP employs our designed Iterative Bilateral Extremum Matching method to prevent the occurrence of some incorrect pseudo-labels. Thirdly, we propose a Scene Prior Contrastive Learning module, which encourages the model to directly acquire the scene prior knowledge from the target domain, thereby mitigating the generation of noise. By suppressing noise contamination, avoiding noise occurrence and mitigating noise generation, our framework achieves state-of-the-art performance on two benchmark datasets, PRW with 50.2% mAP and CUHK-SYSU with 87.0% mAP.

## 1 Introduction

Person search aims to localize and identify a query person from a gallery of scene images. It can be taken as a joint task of person detection and re-identification (re-id) in an end-to-end manner, where supervised learning has made significant advancements. However, notable performance degradation is observed in these methods when deployed to new application scenarios, primarily attributed to domain gaps induced by factors such as camera configuration. Additionally, the process of annotating an adequate amount of training data for a particular domain is both arduous and costly. Consequently, unsupervised domain adaptation (UDA) holds considerable promise in practical real-world scenarios for person search.

UDA in person search aims to achieve model generalization from labeled source data to unlabeled target deployment scenarios. DAPS [11], a pioneering work in applying UDA to person search, employs implicit domain alignment in conjunction with a pseudo-labeling method to effectively bridge the gap between the source and target domains. Building upon DAPS, Almansoori *et al.* proposed DDAM [1], which generates a hybrid domain and learns within it to minimize the distance between the two domains. Although these methods achieve satisfactory performance, they failed to

---

[*]Corresponding Author
[†]Equal Contribution
Our code is available at https://github.com/whbdmu/STCSP.

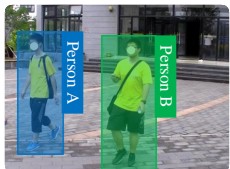 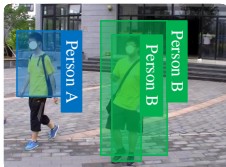 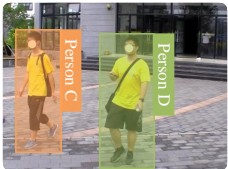 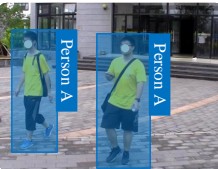

| (a) Correct labeling. | (b) Inaccurate BBoxes. | (c) Incorrect Identities. | (d) Identity Conflict. |

Figure 1: Display of incorrect pseudo-labels. (a) A correct pseudo-label has an accurate BBox and a correct person identity. (b) An inaccurate pseudo-BBox manifests in the form of either encapsulating an erroneous region or exhibiting the overlap of multiple boxes. (c) An incorrect identity label comes from situations where an identity is assigned to different individuals or where different instances of a person are assigned different identities. (d) Assigning the same identity label to intra-scene persons is a clear error.

effectively eliminate the interference of noise stemming from the source-biased models on pseudo-labels. As illustrated in Fig. 1b, the inaccurate bounding boxes (BBoxes) inevitably lead to detection failures, which can induce significant degradation in person search performance. Additionally, as shown in Fig. 1c, the incorrect labels will directly cause confusion in identifying the person identities. Furthermore, the noise accumulates with the increase of epochs, forming a negative feedback mechanism that will cause performance degradation of the model during the later training phase. Therefore, eliminating the interference of the noise on pseudo-labels is a major challenge. In addition, previous studies failed to realize that a scene can provide relationships for its inner persons. Specifically, the identities of persons within a scene image are inherently distinct, and this is referred to as scene prior. The omission of considering and exploiting this scene prior results in a conspicuous error as illustrated in Fig. 1d. Consequently, the utilization of the scene prior is another crucial issue.

To address these issues, we propose a novel Spatiotemporal Consensus with Scene Prior (STCSP) framework that eliminates the impact of the noise through suppressing noise contamination, preventing some erroneous pseudo-labels and mitigating noise generation. Our framework progressively bridges the domain gap via three core innovations. First, we devise a Spatiotemporal Consensus (STC) pipeline to suppress noise propagation in pseudo-labels. STC maintains a memory bank of the previous detection and re-id information. To filter out temporal jitters, STC leverages the temporal consensus between the information in the memory bank and the current state. In addition, for the detection and re-id subtasks respectively, STC exploits the spatial consensus in the scene space of the image and the latent space of clustering to eliminate spatial outliers. Second, leveraging the intrinsic scene prior, STCSP conducts bipartite matching for the instances in any two of all the images, thus averting the error depicted in Fig. 1d. However, deriving the optimal solutions for bipartite matching across thousands of images is computationally prohibitive. To tackle this challenge, we propose an Iterative Bilateral Extremum Matching (IBEM) method, leveraging GPU acceleration, to seek the approximate solutions. Third, we propose a Scene Prior Contrastive Learning (SPCL) module that encourages the model to learn discriminative person features within the target domain. Specifically, besides proxy learning, SPCL steers the model to learn identity heterogeneity within the same scene, enabling the model to directly absorb knowledge from the target domain. The existence of this knowledge effectively reduces the noise output of the model.

Our major contributions are summarized below:

- A Spatiotemporal Consensus pipeline is proposed to suppress the noise origination from the domain gap, thereby generating reliable pseudo-labels.

- An Iterative Bilateral Extremum Matching method is designed. Within a few seconds, it can match the instances in any two of all thousands of images.

- A Scene Prior Contrastive Learning module is proposed, which encourages the model to directly acquire knowledge from the target domain.

With the spatiotemporal consensus and the scene prior, STCSP achieves state-of-the-art (SOTA) performance on two benchmark datasets. Specifically, it attains a mean average precision of 50.2% on the PRW dataset and 87.0% on the CUHK-SYSU dataset.

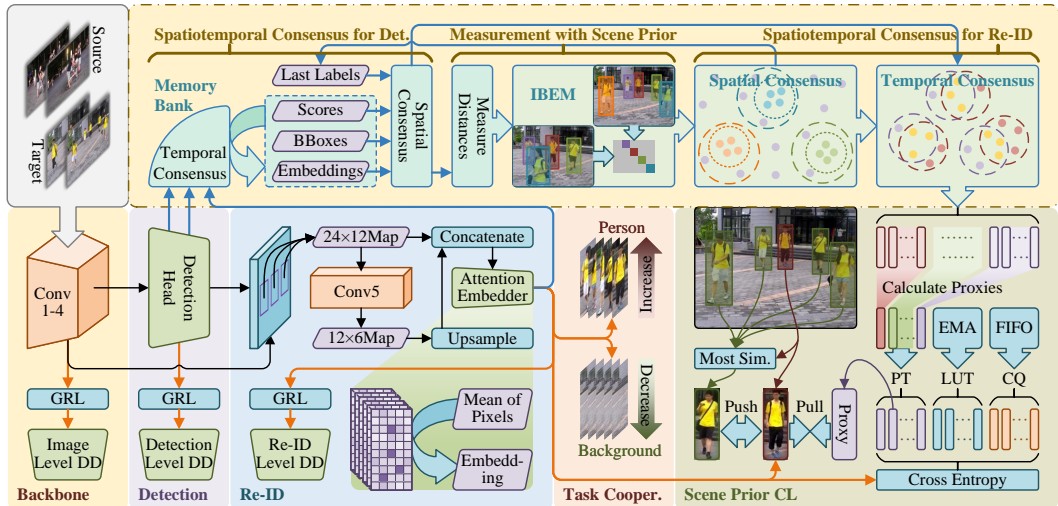

Figure 2: Architecture of the proposed framework. This figure is structured into two rows. The first row presents the input images and the spatiotemporal consensus pipeline. The second row, arranged from left to right, encompasses the backbone network, detection module, re-id module, task cooperation module, and scene prior contrastive learning module.

## 2 Method

### 2.1 Framework Overview

Our UDA person search model is an end-to-end architecture based on SeqNet [12]. Differently, we replaced the backbone network and person feature embedder with ConvNeXt-B [16] and our designed attention embedder, respectively. Following the approach of DAPS [11], we introduce gradient reversal layers (GRLs) and domain discriminators (DDs) for three level features. We adhere to the principle of "sample quality first" and use our STC pipeline to obtain reliable pseudo-labels. The SPCL and task cooperation (TC) modules are employed to steer our model to learn discriminative features.

Prior to the training of each epoch, the STC pipeline preprocesses the unlabeled target domain. As in the first row of Fig. 2, the temporal consensus updates the memory bank with model-inferred BBox scores, BBoxes, and embeddings. The spatial consensus then screens out pseudo-BBoxes from the memory bank. Subsequently, their embeddings are calculated by Re-Ranking [30] to form a cross-distance matrix. This matrix undergoes bipartite matching via the IBEM method, and the result is used for matrix sparsity regularization. The regularized matrix generates spatial consensus clusters, which combine with those from the previous epoch to form spatiotemporal ones. The center of each cluster serves as a proxy sent to the SPCL.

During the training phase, the model is fed source and target domain mini-batches in an alternating sequence, with parameters being updated every two mini-batches. As shown in the second row of Fig. 2, the first four ConvNeXt-B [16] stages are responsible for extracting scene features. The feature map is then fed into the Faster-RCNN-based [20] detection module for BBox generation, where RoI-Align pools the instance feature map for each BBox and the Attention Embedder encodes it into an embedding. Finally, the embedding is sent to the TC and SPCL modules for loss calculation.

### 2.2 Spatiotemporal Consensus for Detection

**Temporal Consensus for Detection.** For the estimated BBoxes $\mathcal{H}$ inferred by the model and the anchor BBoxes $\mathcal{G}$ stored in the memory bank, we use the intersection-over-union (IoU) scores as the basis for the matching between them. The unmatched BBoxes and their accompanying BBox scores and instance embeddings in the memory bank will be removed, and the remaining contents will be

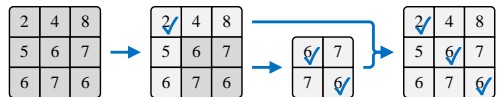

Figure 3: An example of IBEM. On the left and right are the cost matrix and the optimal solution of a minimum bipartite matching task, respectively. The first BEM extracts the primary pair, and the secondary BEM resolves the residuals.

updated with momentum $\eta$. The matching and updating operations are expressed as follows:

$$\widetilde{\mathcal{G}} = \left\{ \eta g + (1-\eta)h \left| \begin{array}{l} g \in \mathcal{G}, \ h \in \mathcal{H}, \ \rho(g,h) > \theta, \\ \underset{a \in \mathcal{G}}{\operatorname{argmax}} \, \rho(h,a) = g, \ \underset{b \in \mathcal{H}}{\operatorname{argmax}} \, \rho(g,b) = h \end{array} \right. \right\}, \quad (1)$$

where $\widetilde{\mathcal{G}}$ is a set of updated anchor BBoxes, $\rho$ is a box IoU function and $\theta$ is a threshold for the IoU score. In this equation, the anchor BBox $g$ and the estimated BBox $h$ are mutually IoU-optimal, and the score also meets the threshold. Matching the anchor with the maximum IoU for each estimate is to avoid updating multiple anchors for one estimate. Based on the assumption that the pseudo-labels are true, we consider each anchor to be true. Therefore, selecting the estimate with the maximum IoU is equivalent to maximizing the likelihood probability of generating that anchor.

In addition, the estimated BBoxes in $\mathcal{H}$ that are independent of all anchor BBoxes are used to generate new anchors. Whose definition is as follows:

$$\widehat{\mathcal{H}} = \left\{ h \in \mathcal{H} \left| \max_{g \in \mathcal{G}} \rho(g,h) \leq \theta \right. \right\}. \quad (2)$$

The boxes in $\widehat{\mathcal{H}}$ with a high IoU score between them will be merged to ensure the independence among the new anchors. First, a variant of non-maximum suppression (NMS) operation with threshold $\theta$ is applied to divide $\widehat{\mathcal{H}}$ into multiple subsets, as follows:

$$\widehat{\mathcal{H}} \xrightarrow{\text{Split by NMS}_\theta} \left\{ \widehat{\mathcal{H}}_1, \widehat{\mathcal{H}}_2, \cdots \right\}. \quad (3)$$

Specifically, a box retained by the NMS and the boxes it suppresses are grouped together. Finally, the mean values of each subset from the new anchors $\widehat{\mathcal{G}}$. Notably, the score and embedding corresponding to each BBox will undergo the same operations during this process. Each update of the memory bank is a deep integration of historical and current knowledge. Therefore, the result $\mathcal{G} \leftarrow \widetilde{\mathcal{G}} \cup \widehat{\mathcal{G}}$ is a temporal consensus.

**Spatial Consensus for Detection.** The comprehensive analysis of Eqs. (1) to (3) reveals that the maximum value of the IoU scores among the anchor BBoxes in the memory bank is approximately $\theta$. This indicates that there may be multiple BBoxes stacked on one person instance, as shown in the right of Fig. 1b. Therefore, we apply a NMS with threshold $\varphi$ to screen out reliable BBoxes from the memory bank, thereby establishing spatial consensus on each image.

### 2.3 Measurement with Scene Prior

Following previous works, we adopt Re-Ranking [30] method to compute pairwise distances between instance embeddings. To optimize computational performance, we have developed a GPU-accelerated implementation using PyTorch. However, different from previous works, we find that scenes prior can sparsely regularize the distance matrix, thus avoiding the error as shown in Fig. 1d. Specifically, our approach constructs inter-scene cost matrices from the instance distances, performing minimum bipartite matching to set unmapped pair distances to infinity, thereby enforcing intra-scene identity uniqueness. However, the $O\left(n^3\right)$ omplexity of exact solvers like Hungarian algorithm [10] becomes prohibitive at scale ($\approx$ 2hours/10k images).

**Iterative Bilateral Extremum Matching.** To relieve this dilemma, we propose an IBEM method to calculate the approximate solution. Specifically, as shown in algorithm 1, the IBEM iteratively identifies entries that are extrema (*e.g.*, minima or maxima) in both their row and column, then removes the matched entries' rows and columns for further iterations until the conditions are met.

---

**Algorithm 1** Iterative Bilateral Extremum Matching

---

**Require:** Cost matrix $\mathcal{C} = \{c_{1,1}, \cdots\}$ and its row and column indices $I_1$ and $J_1$.
**Ensure:** Matching pairs: $\mathcal{M} = \mathcal{M}_1 \cup \mathcal{M}_2 \cup \cdots \cup \mathcal{M}_\delta$
  **while** $t = 1$ to $\delta$, $|I_t| \times |J_t| \,/\, |I_1| \times |J_1| < \vartheta$ **do**

$$\mathcal{M}_t \leftarrow \left\{ (i,j) \middle| c_{i,j} = \min_{k \in I_t} c_{k,j}, c_{i,j} = \min_{k \in J_t} c_{i,k} \right\}$$

$$I_{t+1} \leftarrow \{i \in I_t | i \neq p, (p,q) \in \mathcal{M}_t\}$$
$$J_{t+1} \leftarrow \{j \in J_t | j \neq q, (p,q) \in \mathcal{M}_t\}$$

  **end while**

---

Although some rows and columns may remain unmatched during each iteration of bilateral extremum matching (BEM), we rigorously prove by contradiction that a scenario with complete unmatching of all rows and columns is theoretically impossible. Consequently, full instance matching is guaranteed via the iterative process. To balance computational efficiency and convergence speed, we introduce two hyperparameters, an unmatched proportion threshold $\vartheta$ and a maximum number of iterations $\delta$ as limitations. Fig. 3 shows an example of this method. Notably, by leveraging GPU acceleration, IBEM completes cross-image matching across thousands of instances within seconds.

### 2.4 Spatiotemporal Consensus for Re-ID

**Spatial Consensus for Re-ID.** Conceptually, a discriminative cluster requires sufficient separation from other instances. However, the uneven density of the latent space renders a fixed-distance threshold inadequate for the specificity evaluation. To address this challenge, we propose a spatial clustering consensus strategy, which identifies reliable clusters through dual-granularity pattern analysis. Specifically, we implement a dual-density DBSCAN strategy: A conservative density with radius $\epsilon$ and min sample size $\kappa$ captures coarse-grained clusters, while an aggressive density with radius $\alpha \cdot \epsilon$ and min sample size $\alpha \cdot \kappa$ detects fine-grained clusters. Then, through the cluster consensus method, our strategy synthesizes spatial consensus clusters, as shown on the first row of Fig. 2. This cross-validation mechanism effectively distinguishes reliable clusters, ensuring robustness against the latent space with uneven density.

**Temporal Consensus for Re-ID.** As the instances iteratively update their latent representations, the temporally stable clusters maintain structural cohesion despite positional shifts. Therefore, to screen out the clusters with temporal robustness, we propose a temporal clustering consensus strategy, which integrates the spatial consensus results across previous and current epochs. This cross-epoch fusion process identifies the persistent clusters by the cluster consensus method, thereby forming temporal consensus results. Specifically, as shown on the first row of Fig. 2, the temporal consensus window is constrained to two adjacent epochs, enforcing gradual cluster evolution while preserving temporal consensus across training iterations.

**Cluster Consensus.** For a certain element $\xi$ in a set $\Xi = \{\xi_1, \xi_2, \cdots, \xi_n\}$ containing $n$ instances, there are two different strategies, $\Phi$ and $\Psi$, to obtain its cluster labels $\Phi(\xi, \Xi)$ and $\Psi(\xi, \Xi)$ respectively. So, in strategy $\Phi$, the cluster which $\xi$ belongs to is as follows:

$$\mathrm{F}(\xi, \Xi, \Phi) = \{x \in \Xi | \Phi(x, \Xi) = \Phi(\xi, \Xi)\}. \tag{4}$$

Based on F, an intersection-over-union score of the clusters where $\xi$ is located in $\Phi$ and $\Psi$ can be calculated as follows:

$$\mathrm{f}(\xi, \Xi, \Phi, \Psi) = \frac{|\mathrm{F}(\xi, \Xi, \Phi) \cap \mathrm{F}(\xi, \Xi, \Psi)|}{|\mathrm{F}(\xi, \Xi, \Phi) \cup \mathrm{F}(\xi, \Xi, \Psi)|}, \tag{5}$$

where $\mathrm{F}(\xi, \Xi, \Phi)$ and $\mathrm{F}(\xi, \Xi, \Psi)$ represent the clusters where $\xi$ is placed in $\Phi$ and $\Phi$, respectively. Then, the maximum IoU score among all instances in $\mathrm{F}(\xi, \Xi, \Phi)$ is as follows:

$$\mathrm{g}(\xi, \Xi, \Phi, \Psi) = \max_{x \in \mathrm{F}(\xi, \Xi, \Phi)} \mathrm{f}(x, \Xi, \Phi, \Psi). \tag{6}$$

Finally, a cluster consensus label for $\xi$ is obtained through the following conditional mapping:

$$\mathrm{h}(\xi, \Xi, \Phi, \Psi) = \begin{cases} \Phi(\xi, \Xi), & \mathrm{f}(\xi, \Xi, \Phi, \Psi) = \mathrm{g}(\xi, \Xi, \Phi, \Psi) > 0.5 \\ -1, & \text{else} \end{cases}, \tag{7}$$

where $f(\xi, \Xi, \Phi, \Psi) = g(\xi, \Xi, \Phi, \Psi)$ means that $\xi$ lies in the intersection of $F(\xi, \Xi, \Phi)$ and $F(\xi, \Xi, \Psi)$, which has the maximum IoU score within $F(\xi, \Xi, \Phi)$. Moreover, we enforce a 0.5 IoU threshold to validate high confidence in $\Phi$ and $\Psi$ consensus. Besides, an instance with cluster label of $-1$ denotes an outlier.

## 2.5 Scene Prior Contrastive Learning

Existing works have achieved satisfactory results through contrastive learning using person feature proxies. However, they failed to realize and utilize scene prior to guide the model learning. Therefore, we design a triplet loss function to encourage the model to learn this prior knowledge. Specifically, we introduce a person proxy table (PT) $V = \{v_1, v_2, \cdots\}$ for the target domain. Before each training epoch, the PT will be re-formed by taking the mean value of embeddings in each cluster. And, during the training, it is updated online with a momentum $\lambda$. Given a set $\mathcal{I}$ of the instance feature embeddings in a scene image, the margin $m$ of an embedding $x$ in $\mathcal{I}$ is calculated as follows:

$$m = \langle v_{\mathrm{L}(x)}, x \rangle - \max_{a \in \mathcal{A}} \langle a, x \rangle, \quad \mathcal{A} = \{a \in \mathcal{I} | \mathrm{L}(a) \neq \mathrm{L}(x)\}, \tag{8}$$

where $\langle \cdot, \cdot \rangle$ denotes the inner product between two embeddings and $\mathrm{L}(\cdot)$ represents taking the label corresponding to the embedding. $v_{\mathrm{L}(x)}$ is the proxy in $V$ corresponding to the $x$ and $\mathcal{A}$ is all the embeddings in $\mathcal{I}$ that have different labels from $x$. Besides, we use a hyper-parameter $M$ to evaluate the margin. And the triplet loss objective is to increase the margin $m$ to $M$:

$$\mathcal{L}_{Triplet} = \max(M - m, 0). \tag{9}$$

In addition, we also employ a cross-entropy loss in this module. It calculates the probability that the label of $x$ is $\mathrm{L}(x)$, and uses cross-entropy to supervised the probability.

$$\mathcal{L}_{CE} = -\log p, \quad p = \frac{\varepsilon\left(v_{\mathrm{L}(x)}, x\right)}{\sum\limits_{a \in V} \varepsilon(a, x) + \sum\limits_{b \in U} \varepsilon(b, x) + \sum\limits_{c \in Q} \varepsilon(c, x)}, \quad \varepsilon(a, b) = \exp\left(\langle a, b \rangle / \tau\right), \tag{10}$$

where $\tau = 1/30$ is hyper-parameter which adjusts the softness of the probability distribution. $U$ and $Q$ are respectively the lookup table (LUT) and circular queue (CQ) proposed by OIM, which are updated by the embeddings generated from the source domain. Finally, the scene prior contrastive learning loss is obtained by adding two terms together:

$$\mathcal{L}_{SPCL} = \mathcal{L}_{Triplet} + \mathcal{L}_{CE}. \tag{11}$$

## 2.6 Task Cooperation

BNR [9] loss utilizes the foreground and background labels from the detection head to supervise the embeddings generated by the re-id head. It can encourage the backbone to enhance the foreground features and suppress the background features, thereby improving the model's detection and re-id performance simultaneously. We improved the equation for calculating the probability that an embedding $x$ belongs to the foreground as follows:

$$p = \sigma\left[\mathrm{BN}\left(\|x\|_2^2 \big/ 2\right)\right], \tag{12}$$

where $\sigma(\cdot)$ is a sigmoid function, $\mathrm{BN}(\cdot)$ is an batch normalization [8] layer, and $\|\xi\|_2$ is the $L_2$-norm value of the feature embedding $x$. We employ the square $L_2$-norm as the regularization term to impose independent constraints on each dimension of $x$ and prevent outliers in some dimensions from influencing others through cross-term interactions. And dividing by 2 is to simplify the derivative operation. Finally, Focal Loss [14] is calculated for $p$, which focus training on the hard negatives:

$$\mathcal{L}_{TC} = -(1 - p_t)^2 \log(p_t), \quad p_t = \begin{cases} p, & y = 1 \\ 1 - p, & y = 0 \end{cases}, \tag{13}$$

where $y$ is a label (either 0 or 1) indicating whether its corresponding embedding is marked as a background or person.

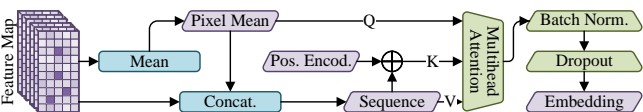

Figure 4: Details of the proposed attention embedder. "Pos. Encod.", "Concat." and "Batch Norm." respectively represent "Positional Encodings", "Concatenate" and "Batch Normalization".

### 2.7 Attention Embedder

Inspired by the attention pool module in CLIP [19], we propose a novel attention embedder as shown in Fig. 4. Distinct from the attention pool module, our design introduces two critical enhancements to the multi-head attention output: Batch Normalization [8] (BN) and Dropout [21]. BN standardizes the distribution of each embedding dimension by centering it near zero with unit variance, mitigating its congregation in positive or negative ranges. This normalization encourages a discriminative embedding and enhance its expressivity in the latent space. Subsequently, Dropout injects stochasticity during training, preventing over-reliance on specific embedding dimensions. Through the synergism of these components, a discriminative and robust embedding is obtained.

## 3 Experiments

### 3.1 Experimental Setup and Implementation Details

**CUHK-SYSU** [25] is a large-scale person search dataset that contains 18,184 images collected from handheld cameras, movies, and TV shows, resulting in significant scene diversity. It encompasses 8,432 unique person identities and 96,143 annotated bounding boxes. The dataset is split into a training set with 5,532 identities and 11,206 images, and a test set with 2,900 queries and 6,978 gallery images. For each query, the dataset defines a gallery size ranging from 50 to 4,000, with a default gallery size of 100 images.

**PRW** [29] is composed of video frames captured by six fixed cameras on a university campus. It contains 11,816 scene images with 932 distinct person identities and 43,110 annotated bounding boxes. The training set consists of 483 identities with 5,704 images, and the test set contains 2,057 queries and 6,112 scene images. For each query, the dataset uses all images in the test set except for the query as the gallery.

**Evaluation Metric.** Following the established setting in previous work, the mean Average Precision (mAP) and top-1 accuracy (top-1) are employed to evaluate the performance for person search.

**Implementation Details.** We conduct all experiments on a NVIDIA A800 GPU and implement our model with PyTorch [18]. ConvNeXt [16] pre-trained on ImageNet [5] is adopted as the backbone network. During training, we set the batch size to 4, and use Automatic Mixed Precision (AMP). Adam is used to optimize our model for 20 epochs with an initial learning rate of 0.0001 and a weight decay of 0.01, which is warmed up during the first epoch and reduced by a factor of 0.1 at epochs 8 and 14. When PRW is used as the target domain, our model undergoes pre-training on the source domain CUHK-SYSU for 5 epochs before commencing joint training. Conversely, it pre-train on PRW for 3 epoch. The thresholds $\theta$ and $\varphi$ in Sec. 2.2 are set to 0.7 and 0.4, respectively. For algorithm 1, the limitations $\delta$ and $\vartheta$ are set to 2 and 0.2, respectively. The hyper-parameter $\epsilon$, $\kappa$ and $\alpha$ in Sec. 2.4 is set to 0.5, 4 and 0.8, respectively. For Eq. (9), we initialize the hyper-parameter $M$ as 0.35. For the attention embedder, the probability of the Dropout is set to 0.1.

### 3.2 Comparison to the State-of-the-Arts

**Comparison on PRW.** The 3rd column of Tab. 1 shows the performance of all methods on the PRW test set. As shown in the table, our framework outperforming all the other UDA and weakly-unsupervised methods, whether based on ConvNeXt or ResNet. Especially for the current best-performing DDAM [1], STCSP outperforms it by 13.5% and 6.1% in mAP and top-1 accuracy, respectively. And when both employ ResNet-50 as the backbone, our method still outperform it by 5.7% and 0.9%. Notably, our method outperforms most fully supervised methods, despite they using ground truth labels to train models.

Table 1: Comparison of mAP(%) and top-1(%) accuracy with the fully-supervised, weakly-supervised and UDA methods on the PRW and CUHK-SYSU test sets. For UDA methods, the best and second best scores are shown in bold and underlined, respectively.

| Method | Backbone | PRW | | CUHK-SYSU | |
|---|---|---|---|---|---|
| | | mAP | top-1 | mAP | top-1 |
| *Fully Supervised Learning* | | | | | |
| OIM [25] | ResNet-50 | 21.3 | 49.4 | 75.5 | 78.7 |
| IAN [24] | ResNet-50 | 23.0 | 61.9 | 76.3 | 80.1 |
| NPSM [15] | ResNet-50 | 24.2 | 53.1 | 77.9 | 81.2 |
| RCAA [2] | ResNet-50 | - | - | 79.3 | 81.3 |
| CTXG [28] | ResNet-50 | 33.4 | 73.6 | 84.1 | 86.5 |
| QEEPS [17] | ResNet-50 | 37.1 | 76.7 | 88.9 | 89.1 |
| NAE+ [3] | ResNet-50 | 44.0 | 81.1 | 92.1 | 92.9 |
| AlignPS+ [27] | ResNet-50 | 46.1 | 82.1 | 94.0 | 94.5 |
| SeqNet [12] | ResNet-50 | 46.7 | 83.4 | 93.8 | 94.6 |
| SEAS [9] | ResNet-50 | 52.0 | 85.7 | 96.2 | 97.1 |
| SEAS [9] | ConvNeXt-B | 60.5 | 89.5 | 97.1 | 97.8 |
| *Weakly Supervised Learning* | | | | | |
| R-SiamNet[6] | ResNet-50 | 21.4 | 75.2 | 86.0 | 87.1 |
| CGPS [26] | ResNet-50 | 16.2 | 68.0 | 80.0 | 82.3 |
| SSL [22] | ResNet-50 | 30.7 | 80.6 | 87.4 | 88.5 |
| KCD [13] | ResNet-50 | 40.5 | 83.6 | 86.8 | 88.2 |
| DICL [23] | ResNet-50 | 35.5 | 80.9 | 87.4 | 88.8 |
| *Unsupervised Domain Adaptation* | | | | | |
| DAPS [11] | ResNet-50 | 34.7 | 80.6 | 77.6 | 79.6 |
| FOUS [4] | ResNet-50 | 35.4 | 80.8 | 78.7 | 80.5 |
| DDAM [1] | ResNet-50 | 36.7 | 81.2 | 79.5 | 81.3 |
| **STCSP (ours)** | ResNet-50 | 42.4 | 82.1 | 80.4 | 82.5 |
| **STCSP (ours)** | ConvNeXt-B | **50.2** | **87.3** | **87.0** | **87.9** |

(a) Comparison of mAP.

(b) Comparison of top-1 accuracy.

Figure 5: Comparison of mAP and top-1 accuracy on CUHK-SYSU across various gallery sizes.

**Comparison on CUHK-SYSU.** The performance of our method and the other methods on the CUHK-SYSU test set is shown in the 4th column of Tab. 1. STCSP based on ConvNeXt or ResNet outperforms all the other UDA methods. It achieves the best scores of 87.0% in mAP and 87.9% in top-1 accuracy, surpassing DDAM [1] by 7.5% and 6.6% in mAP and top-1 accuracy, respectively. The CUHK-SYSU training set boasts greater diversity and a larger quantity of scene images compared to the PRW training set. As a result, UDA from PRW to CUHK-SYSU presents a formidable challenge. Despite this, our method still outperforms several weakly-supervised and fully-supervised methods on the CUHK-SYSU test set.

We further perform a comparison between our STCSP and other methods on the CUHK-SYSU test set with gallery sizes ranging from 50 to 4,000. Fig. 5 shows the performance curve of all the methods in terms of mAP and top-1 as the gallery size increases. Since it is a challenge for all compared methods to consider more distracting persons in the gallery set, the performance of them is reduced as the gallery size increases. However, our method consistently outperforms all the other methods in

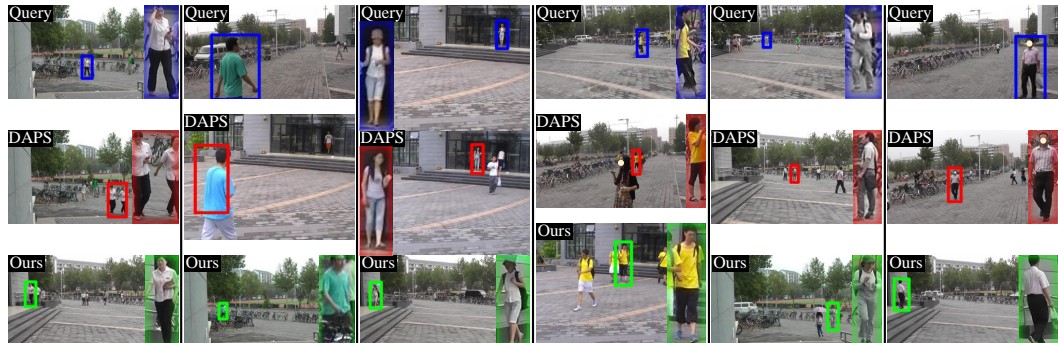

Figure 6: Qualitative comparison of our method with DAPS on the PRW test set. Each column represents a comparison result. The blue bounding boxes denote the queries, while the green and red bounding boxes denote correct and incorrect top-1 matches, respectively.

Table 2: Comparative results of the different strategies for our framework.

| Strategy | PRW | | CUHK-SYSU | |
|---|---|---|---|---|
| | mAP | top-1 | mAP | top-1 |
| *w/o* Detection Spatial Consensus | 46.8 | 85.7 | 85.4 | 85.4 |
| *w/o* Detection Temporal Consensus | 47.7 | 86.0 | 86.4 | 87.5 |
| *w/o* Re-ID Spatial Consensus | 46.9 | 85.8 | 86.7 | 87.7 |
| *w/o* Re-ID Temporal Consensus | 49.0 | 87.4 | 86.6 | 87.5 |
| *w/o* IBEM | 48.2 | 85.9 | 85.9 | 86.6 |
| *w/o* Triple Loss of SPCL | 47.9 | 87.0 | 86.2 | 87.4 |
| | **50.2** | **87.3** | **87.0** | **87.9** |

mAP and top-1, whether based on ConvNeXt or ResNet. This indicates that our method has excellent generalization ability.

**Qualitative Results.** Some example person search results are illustrated in Fig. 6. The 5th column of the figure shows the result of using a low-resolution image as the query. When multiple highly similar characters appear in a scene, the search results are illustrated in the 1st and 4th columns. The 2nd column shows the result of the person's body being obstructed. The 3rd and 6th columns show the result when the scene and viewpoint change. It can be observed that our successfully handles low-resolution, similar-character, occlusion, cross-scene and viewpoint variation, while other state-of-the-art methods such as DAPS fail in these scenarios. This demonstrates the robustness of our method.

### 3.3 Ablation Study

**Spatiotemporal Consensus for Detection.** To verify the effectiveness of our STC strategy for detection, we evaluate separately the performance of our framework in two scenarios: detection without spatial consensus and without temporal consensus, and report the results in Tab. 2. We find that STCSP exhibits significant performance degradation when lacking the spatial or temporal consensus for detection. This results indicate that the STC strategy for detection significantly elevates the quality of pseudo-BBoxes.

**Spatiotemporal Consensus for Re-ID.** Tab. 2 shows the performance of our framework being employed with different strategies on the two benchmarks. First. to validate the effectiveness of the spatial consensus, we replace it with a DBSCAN method. The results reveal that our strategy of employing a loose and compact clustering space to form spatial consensus can generate reliable clusters. Furthermore, The results after removal of the temporal consensus are also reported in Tab. 2. Through comparison, we can see that the temporal consensus improve the performance of our model

Table 3: Comparative results of the different components for our framework. RN, CN, NAE and AE indicate ResNet-50, ConvNeXt-B, Norm-Aware Embedding and Attention Embedder, respectively.

| Backbone | | Embedder | | PRW | | CUHK-SYSU | |
|---|---|---|---|---|---|---|---|
| RN | CN | NAE | AE | mAP | top-1 | mAP | top-1 |
| ✓ | ✗ | ✓ | ✗ | 41.1 | 82.2 | 80.3 | 82.3 |
| ✓ | ✗ | ✗ | ✓ | 42.4 | 82.2 | 80.4 | 82.5 |
| ✗ | ✓ | ✓ | ✗ | 45.8 | 85.6 | 85.6 | 85.5 |
| ✗ | ✓ | ✗ | ✓ | **50.2** | **87.3** | **87.0** | **87.9** |

on both of PRW and CUHK-SYSU. We attribute these improvements to the stable clusters formed by the temporal consensus.

**Scene Prior.** We evaluated the influence exerted by the lack of scene priors within the related components on the performance manifestations of our framework, and report the results in Tab. 2. As presented in the 5th row of the table, when the IBEM is absent from our framework, entailing the potential occurrence of the error depicted in Fig. 1d, the performance of our framework experiences a substantial decline across both datasets. Notably, this decline is more pronounced on the PRW dataset. Similarly, as indicated in the 6th row of the table, a comparable scenario transpires when the triplet loss within the SPCL is lacking. The significant performance degradation observed on PRW can be attributed to the existence of persons with a high similarity within some scene images in this dataset. This indicates that scene prior effectively steers the model to learn discriminative features.

**Model Architecture.** As shown in Tab. 3, in order to assess the influence of diverse model architectures on our framework, we report the performance outcomes under all cross-combination modes of ResNet-50 (RN) [7] and ConvNeXt-B (CN) [16] with Norm-Aware Embedding (NAE) [3] and Attention Embedder (AE). When comparing backbone networks, ConvNeXt-B significantly outperforms ResNet-50 across all configurations. Furthermore, the AE leads to a further performance improvement compared to the NAE, especially when it is integrated with CN. Notably, the performance gains of AE are relatively less pronounced when combined with RN, suggesting that the compatibility between the backbone network and the embedder affects the performance of the framework. The results indicate that the combination of CN and AE is the optimal configuration.

## 4 Conclusion

In this paper, we focus on UDA Person Search, where existing methods suffer from noise accumulation caused by source-biased pseudo-labels. To bridge the domain gap, we propose the STCSP framework, which systematically suppresses noise contamination, avoids noise occurrence, and mitigates noise generation. With STC pipeline, IBEM method and SPCL module, our framework achieves SOTA performance. For future research, the IBEM algorithm can be extended to other tasks that demand the rapid completion of extensive bipartite matchings.

## Acknowledgments and Disclosure of Funding

This work was supported in part by the National Natural Science Foundation of China Grant 62576067 and 62501226, National Key Research and Development Program of China Grant 2024YFB4710800, Liaoning Provincial Natural Science Foundation Grant 2025-YQ-01 and 2024-MS-012, Dalian Science and Technology Talent Innovation Support Plan Grant 2024RY010, Natural Science Foundation of Hebei Province Grant F2025201037.

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
