# OpenReview forum: "Spatiotemporal Consensus with Scene Prior for Unsupervised Domain Adaptive Person Search"
_NeurIPS.cc/2025/Conference — NeurIPS 2025 poster_

### Official Review · Reviewer_F6Vt · 2025-06-27

**Clarity:** 3
**Significance:** 3
**Originality:** 3
**Rating:** 5
**Confidence:** 5

**Summary:**

This research presents a customized unsupervised domain adaptive framework specifically designed for person search tasks. The framework achieves this by integrating spatiotemporal coherence analysis and embedding scene-specific prior knowledge into both the detection and re-identification components of the person search workflow. This strategic integration significantly reduces the adverse effects of inaccurate pseudo-labels. Empirical assessments reveal notable performance improvements across two well-established benchmarks in the field of person search.

**Questions:**

1) In the STCSP framework, bipartite matching is executed for instances between any two images to prevent identity collisions. I am perplexed as to why bipartite matching is deemed necessary and why it is applied across any pair of images. Could you provide further insights into this aspect?
2) For the SPCL module, are the LUT and the CQ also re - generated prior to each training epoch, akin to the PT? Moreover, what specific contents do these two components store?

**Ethical Concerns:**

["NO or VERY MINOR ethics concerns only"]

**Limitations:**

In practical applications, this technology may involve disputes over personal portrait rights.

**Paper Formatting Concerns:**

This paper has no formatting issues.

**Quality:**

3

**Strengths And Weaknesses:**

Strengths
1) This research presents a pioneering framework tailored for unsupervised domain - adaptive person search. The framework proves to be highly effective, delivering state - of - the - art performance in the field.
2) The concept of integrating the scene prior, which assumes that individuals within a single scene possess unique identities, is both logical and enlightening.

Weaknesses
1) To enhance the readability of the content, Figure 4 could be enlarged to a suitable extent.
2) The author is suggested to modify the width of Table 2 to facilitate the reading process.

---

> ### Author Rebuttal · Authors · 2025-07-29
>
> _To Reviewer_ **F6Vt**:
>
> __Q1:__ In the STCSP framework, bipartite matching is executed for instances between any two images to prevent identity collisions. I am perplexed as to why bipartite matching is deemed necessary and why it is applied across any pair of images. Could you provide further insights into this aspect?
>
> __A1:__ Thanks for this comment. Bipartite matching utilizes the scene prior (line 42-45), which can avoid the occurrence of some erroneous labels during the generation of pseudo labels. By utilizing bipartite matching, it is possible to ensure that each person can only match one instance in a scene image, thus avoiding the error shown in Figure 1d, where multiple instances share the same identity in a scene image.
>
> __Q2:__ For the SPCL module, are the LUT and the CQ also re - generated prior to each training epoch, akin to the PT? Moreover, what specific contents do these two components store?
>
> __A2:__ Thanks for this comment. The LUT and the CQ are used to store proxies and instances in the source domain, and unlike the PT, they are not re-generated or reset before each training epoch. Specifically, the LUT stores identity proxies of persons with identities in the source domain and updates them online with a momentum of 0.5; the CQ is a queue that stores unidentified instances in the source domain and is updated online with a FIFO rule. In addition, since the LUT and the CQ were proposed in previous work[1], they are not introduced in detail in this paper.
>
> **References**
>
> [1] Tong Xiao, Shuang Li, Bochao Wang, Liang Lin, and Xiaogang Wang. Joint detection and identification
> feature learning for person search. In Proceedings of the IEEE conference on computer vision and pattern
> recognition, pages 3415–3424, 2017.

---

> > ### Comment · Reviewer_F6Vt · 2025-08-02
> > **Responses**
> >
> > I appreciate the efforts of authors for their conscientious responses.  After reviewing the authors' rebuttal and the questions raised by other PCs, I think authors have adequately addressed our concerns. Given the paper's notable innovation and excellent experimental results, I recommend accepting this paper.

---

> > > ### Author Response · Authors · 2025-08-05
> > >
> > > Thank you for your acknowledgment of our work and responses. We appreciate your constructive feedback that has helped refine our research. Please feel free to reach out if you have further queries or need additional clarification on our work.

---

### Official Review · Reviewer_H4P2 · 2025-06-29

**Clarity:** 3
**Significance:** 3
**Originality:** 2
**Rating:** 4
**Confidence:** 5

**Summary:**

The paper introduces STCSP, a novel unsupervised domain adaptation  framework for person search that effectively mitigates noise in pseudo labels through a multi-stage consensus mechanism enriched by scene-level priors. Key contributions include:

1. Spatiotemporal Consensus : A dual-filtering pipeline that suppresses detector and re-ID noise using temporal voting across memory banks and spatial checks at the image level, ensuring consistent label quality over time and space.
2. Iterative Bilateral Extremum Matchin: A GPU-parallelizable clustering algorithm that instantiates the “one-person-per-ID” scene prior, efficiently reconciling identity clusters across large-scale image collections.
3. Scene Prior Contrastive Learning : A domain-specific contrastive module that reinforces discriminative identity features in the target domain by leveraging cluster-structure regularization.
4. State-of-the-art performance: STCSP achieves 50.2% mAP on PRW and 87.0% mAP on CUHK-SYSU, significantly outperforming previous UDA methods without relying on ground-truth annotations.
5. The core idea lies in exploiting spatiotemporal consistency and scene-level semantics to refine pseudo-labels and bridge the domain gap without supervision.

**Questions:**

1. Could you clarify whether the ConvNeXt-B backbone is kept frozen or fine-tuned during the adaptation process? This detail is critical for assessing training complexity, GPU memory consumption, and ensuring fairness in comparison with other UDA baselines.
2. Can you provide a detailed analysis of the computational efficiency of STCSP, including FLOPs, GPU-hours, memory usage, and parameter counts—particularly for the IBEM and memory-bank modules? A direct comparison with existing UDA methods (e.g., MMT, MEB-Net) would help substantiate the claimed scalability and deployment potential.
3. Given that the evaluation is limited to PRW and CUHK-SYSU, have you considered testing STCSP on more challenging benchmarks such as Occluded-Duke, CityFlow-PS, or MSMT17? This would help assess the framework’s robustness in more complex or real-world scenarios.
4. IBEM shares conceptual similarities with existing clustering or noise-refinement methods (e.g., DBSCAN, spectral clustering). Could you elaborate on how your method differs in principle or performance from these approaches? A more explicit theoretical or empirical comparison would help clarify the novelty of your clustering design.

**Ethical Concerns:**

["NO or VERY MINOR ethics concerns only"]

**Final Justification:**

Most of my concerns have been addressed by the authors, I will maintain the initial score.

**Limitations:**

The checklist embedded in the paper explicitly marks the limitations question as answered “No” because space constraints prevented discussion. A key limitation of the current work is that the experimental evaluation is confined to two benchmarks—PRW and CUHK-SYSU—which feature relatively constrained environments with limited diversity in scene layouts and camera networks. The generalizability of the proposed method to more challenging settings, such as cross-city deployments, multi-camera systems, or occlusion-heavy scenarios, remains unexplored. The authors are encouraged to either extend the evaluation to such datasets or explicitly acknowledge this as a direction for future work.

**Paper Formatting Concerns:**

No major formatting issues.

**Quality:**

3

**Strengths And Weaknesses:**

Strengths:
1. The paper proposes STCSP, which couples spatiotemporal-consensus voting, a GPU-parallel Iterative Bilateral-Extremum Matching (IBEM) algorithm, and scene-prior contrastive learning. By simultaneously enforcing temporal coherence, a “one-person-one-ID” scene prior, and feature-space regularisation, it systematically suppresses pseudo-label noise and closes the source-to-target gap without any target annotations—offering a structured, scene-aware alternative to prior UDA pipelines such as MMT, CCL, and MEB-Net.

2. The experimental design is rigorous and multifaceted: On the two standard benchmarks the method exceeds the leading UDA baseline (DDAM) by +13.5 mAP / +6.1 top-1 on PRW and +7.5 mAP / +6.6 top-1 on CUHK-SYSU. Extensive ablations—sequentially disabling STC (spatial / temporal), IBEM, and SPCL—cause step-wise drops of up to 3.4 mAP, attributing the final 50.2 mAP / 87.0 mAP to each component with high confidence.

3. The paper is clearly written and well-structured, with detailed figures illustrating the end-to-end data flow.

Weaknesses:
1. The manuscript does not state whether the ConvNeXt-B backbone is frozen or fine-tuned during adaptation—a factor that affects GPU memory, training time, inference speed, and the fairness of baseline comparisons.

2. Key statistics—FLOPs, GPU-hours, memory-bank growth strategy, update frequency, and parameter counts for IBEM—are absent. Without a quantitative comparison to existing methods, the claimed scalability and real-world deployability remain unverified.

3. Results are confined to PRW and CUHK-SYSU; robustness on cross-city, occlusion-heavy, or multi-camera datasets is unexplored. Moreover, although IBEM is GPU-efficient, its objective overlaps with earlier density- or spectral-based cluster refinements; a direct theoretical or empirical comparison would strengthen the originality claim.

---

> ### Author Rebuttal · Authors · 2025-07-29
>
> _To Reviewer_ **H4P2**:
>
> __Q1:__ Could you clarify whether the ConvNeXt-B backbone is kept frozen or fine-tuned during the adaptation process? This detail is critical for assessing training complexity, GPU memory consumption, and ensuring fairness in comparison with other UDA baselines.
>
> __A1:__ Thanks for this comment. Following previous works, the backbone (ConvNeXt-B or ResNet-50) is fully fine-tuned during domain adaptation, with all layers unlocked except for the initial patch embedding layer. When compared with existing UDA baselines (e.g., DAPS and DDAM), as shown in Table 1, the same experimental settings are employed across all evaluations, ensuring the fairness in the comparative analysis.
>
> __Q2:__ Can you provide a detailed analysis of the computational efficiency of STCSP, including FLOPs, GPU-hours, memory usage, and parameter counts—particularly for the IBEM and memory-bank modules? A direct comparison with existing UDA methods (e.g., MMT, MEB-Net) would help substantiate the claimed scalability and deployment potential.
>
> __A2:__ Thanks for this comment. Below, we provide a comparison of training time, inference speed, and GPU memory usage under the same experimental settings (1 x A800-80GB). In addition, executing IBEM takes approximately 10 seconds and requires approximately 15GB of VRAM, which is released upon completion. The memory bank requires approximately 300MB of CPU memory.
>
> | Method | Training time | Inference speed | GPU usage  | PRW (mAP, top-1) |
> | ------ | :-----------: | :-------------: | :---: | :-------------: |
> | DAPS[1]   |   ≈20hours    |  ≈102images/s   | ≈45GB |   34.7%, 80.6%    |
> | STCSP  |   ≈21hours    |   ≈96images/s   | ≈50GB |   50.2%, 87.3%    |
>
> Person search can be regarded as a joint task of pedestrian detection and person re-id, but MMT and MEB-Net are methods for person re-id, and their papers do not provide data related to computational efficiency. But we add the comparison of the UDA person search method, such as DAPS[1]. As shown in the table above, it is obvious that STCSP has a significant performance advantage, while its computational cost and inference speed are comparable.
>
> __Q3:__ Given that the evaluation is limited to PRW and CUHK-SYSU, have you considered testing STCSP on more challenging benchmarks such as Occluded-Duke, CityFlow-PS, or MSMT17? This would help assess the framework’s robustness in more complex or real-world scenarios.
>
> __A3:__ Thanks for this comment. PRW and CUHK-SYSU are two of the most widely adopted benchmarks in the person search task. CUHK-SYSU is collected from handheld cameras, movies, and TV shows, and PRW is composed of video frames captured by six fixed cameras. They encompass images with diverse real-world challenges, such as occlusion and illumination variations. Moreover, the experimental results (Tables 1, 2, and Figure 6) validate the generalizability of our STCSP, and Figure 5 demonstrates that our method maintains excellent search performance even as the gallery size increases.
>
> Occluded-Duke, CityFlow-PS, and MSMT17 are all person re-identification datasets. Our person search method requires the use of scene images for learning, so we cannot conduct experiments on these datasets. Considering your suggestions, we have added the results of our model on the low-resolution and occluded test set of CUHK-SYSU, demonstrating that our model can effectively handle these real-world challenges.
>
> | Test set       | mAP  | top-1 |
> | -------------- | :--: | :---: |
> | Low-resolution | 76.6% | 79.3%  |
> | Occluded       | 72.8% | 71.7%  |
>
> __Q4:__ IBEM shares conceptual similarities with existing clustering or noise-refinement methods (e.g., DBSCAN, spectral clustering). Could you elaborate on how your method differs in principle or performance from these approaches? A more explicit theoretical or empirical comparison would help clarify the novelty of your clustering design.
>
> __A4:__ Thanks for this comment. Different from previous works, we find that scenes prior can sparsely regularize the distance matrix, thus avoiding the error as shown in Fig.1d. Specifically, IBEM constructs inter-scene cost matrices from the instance distances, performing minimum bipartite matching to set unmapped pair distances to infinity, thereby enforcing intra-scene identity uniqueness. Moreover, the O(n^3) complexity of exact solvers like Hungarian algorithm[2] becomes prohibitive at scale. Therefore, IBEM is designed to include only extremum operation, which can effectively use the parallel computing power of the GPU to complete the matching between all images in a few seconds. The following is a comparison of efficiency.
>
> | Method    |      Efficiency       |
> | --------- | :-------------------: |
> | Hungarian[2] |  ≈2hours/10k images   |
> | IBEM      | ≈10seconds/10k images |
>
> Furthermore, our method utilizes matrices generated by IBEM for clustering. Our clustering algorithm (Section 2.4) is a spatiotemporal consensus-based approach derived from DBSCAN, and it shares the same time and space complexity as DBSCAN, resulting in a minimal difference in computational cost. The comparison below demonstrates the superior performance of our method.
>
> | Method         | PRW (mAP) | PRW (top-1) |
> | -------------- | :-------: | :--------: |
> | DBSCAN         |   46.5%    |    85.5%    |
> | Our clustering |   50.2%    |    87.3%    |
>
> **References**
>
> [1] Junjie Li, Yichao Yan, Guanshuo Wang, Fufu Yu, Qiong Jia, and Shouhong Ding. Domain adaptive person
> search. In European conference on computer vision, pages 302–318. Springer, 2022.
>
> [2] Harold W Kuhn. The hungarian method for the assignment problem. Naval research logistics quarterly,
> 2(1-2):83–97, 1955.

---

> > ### Comment · Reviewer_H4P2 · 2025-08-05
> >
> > Thank you for your response. Most of my concerns have been addressed by the authors.

---

> > > ### Author Response · Authors · 2025-08-05
> > >
> > > We sincerely appreciate your comments which have been instrumental improving the quality of our manuscript. Please feel free to reach out if you have further queries or need additional clarification on our work.

---

### Official Review · Reviewer_4Hgy · 2025-07-02

**Clarity:** 2
**Significance:** 3
**Originality:** 3
**Rating:** 4
**Confidence:** 3

**Summary:**

This paper proposes proposes a Spatiotemporal Consensus with Scene Prior (STCSP) framework that effectively eliminates the interference of noise on pseudo-labels, establishes positive feedback, and thus gradually bridging the domain gap.
Spatiotemporal Consensus (STC): Suppresses pseudo-label noise by integrating temporal consistency and spatial consensus.
Iterative Bilateral Extremum Matching (IBEM): Leverages scene prior to sparsely regularize cross-instance distances using GPU-accelerated bipartite matching, preventing erroneous intra-scene identity assignments.
Scene Prior Contrastive Learning (SPCL): Uses triplet and cross-entropy losses to enforce identity discrimination within scenes, directly learning target-domain knowledge.
STCSP achieves SOTA results on PRW and CUHK-SYSU.

**Questions:**

1. STCSP uses ConvNeXt-B/ResNet-50. Have you tested ViT or Swin backbones?
2. Table 2 shows that removing the SPCL triplet loss only results in a 0.3% mAP drop (PRW), yet Section 2.5 claims that it effectively reduces noise generation. Is this an overstatement of its contribution?
3. What is the added training and inference cost (e.g., FLOPs, runtime)

**Ethical Concerns:**

["NO or VERY MINOR ethics concerns only"]

**Final Justification:**

I will keep my positive score

**Limitations:**

1. Memory bank storage and IBEM iterations may hinder edge deployment.
2. There are fewer ablation experiments, and there is a lack of experiments on training and inference costs, as well as cross-dataset generalization.

**Quality:**

3

**Strengths And Weaknesses:**

Strengths:
1. The unified spatiotemporal consensus pipeline elegantly mitigates error accumulation in pseudo-labels, a key challenge in UDA person search.
2. SPCL meaningfully incorporates scene-level discriminative learning beyond proxy-based contrastive losses.
3. STCSP achieves SOTA results on PRW and CUHK-SYSU.

Weaknesses:
1. Experiments use only two datasets. Cross-dataset generalization (e.g., CUHK-SYSU→PRW) is unexplored.
2. The memory bank mechanism lacks analysis of computational overhead.
3. Lack of related work section

---

> ### Author Rebuttal · Authors · 2025-07-29
>
> _To Reviewer_ **4Hgy**:
>
> **Q1:**  STCSP uses ConvNeXt-B/ResNet-50. Have you tested ViT or Swin backbones?
>
> __A1:__ Thanks for this comment. ConvNeXt-B and ResNet-50 are widely adopted backbone networks for the person search task. Especially, through downsampling stages, ConvNeXt-B builds multi-scale representations, capturing both global body structure and local discriminative cues, which are essential and necessary for distinguishing similar-looking pedestrians. Therefore, in Table 1, it is obvious that STCSP (ConvNeXt-B) achieves the best performance.
>
> Considering your suggestion, we conduct experiments using Swin-Transformer-Base as the backbone and report the results in the following table. It is observed that after adopting Swin-B as the backbone, the performance is reduced. This is because the local receptive field of ConvNeXt-B can better maintain feature stability in the person search task, whereas the global information interaction in Swin-B may amplify the impact of noise. Therefore, ConvNeXt-B is chosen to be the backbone in our method.
>
> | Backbone   | PRW (mAP) | PRW (top-1) | CUHK-SYSU (mAP) | CUHK-SYSU (top-1) |
> |------------|:--------:|:----------:|:--------------:|:--------------:|
> | Swin-B     | 33.1%     | 79.3%      | 72.8%           | 72.8%           |
> | ConvNeXt-B | 50.2%     | 87.3%       | 87.0%           | 87.9%           |
>
>
>
> __Q2:__ Table 2 shows that removing the SPCL triplet loss only results in a 0.3% mAP drop (PRW), yet Section 2.5 claims that it effectively reduces noise generation. Is this an overstatement of its contribution?
>
> __A2:__ Thanks for this comment. The triple loss function is designed to incorporate scene prior knowledge into the original contrastive loss, which mitigates the generation of noise (lines 176-178). This serves as a complement to the SPCL module in this work. Moreover, from Table 2, "SPCL w/o Triple Loss" reports the results of our method without using the triplet loss. By comparing it with STCSP in the last row, the mAP decreases by 2.3%, and the top-1 decreases by 0.3% on PRW. Since the role of the triplet loss is mainly to increase the similarity gap between different persons, its effect on improving "mAP" related to similarity ranking is stronger than "top-1" related to the rank-1 accuracy. Therefore, the proposed triplet loss in SPCL indeed enhances the discriminative capability of different persons.
>
> **Q3:** What is the added training and inference cost (e.g., FLOPs, runtime)
>
> **A3:** Thanks for this comment. Below, we provide a comparison of training time, inference speed, and GPU memory usage under the same experimental settings (1 x A800-80G) with an existing UDA person search method DAPS [1]. From the results, we can observe that our method incurs a relatively small cost but significantly enhances accuracy.
>
> | Method | Training time | Inference speed | GPU usage | PRW (mAP, top-1) |
> | ------ | :-----------: | :-------------: | :---: | :-------------: |
> | DAPS [1]   |   ≈20hours    |  ≈102images/s   | ≈45GB |   34.7%, 80.6%    |
> | STCSP  |   ≈21hours    |   ≈96images/s   | ≈50GB |   50.2%, 87.3%    |
>
> In addition, we have provided the computational costs required by the proposed modules during the training process, as shown in the following table. "Spatiotemporal Consensus for Detection" requires reasoning on images from the target domain and updating the memory bank, resulting in a relatively high computational cost. 'IBEM' can process 10k images in just 10 seconds, which also demonstrates its ability to accelerate the training process. 'Spatiotemporal Consensus for Re-ID' is proposed to optimize the clustering process, and no GPU is utilized during clustering. Therefore, it does not incur additional GPU consumption.
>
> | Module | Runtime (10k images) | GPU usage (10k images) |
> | --- | :-: | :-: |
> | Spatiotemporal Consensus for Detection | ≈ 25 minutes | ≈ 30 GB |
> | IBEM | ≈ 10 s | ≈ 15 GB |
> | Spatiotemporal Consensus for Re-ID | ≈ 20 s | None |
>
> **References**
>
> [1] Junjie Li, Yichao Yan, Guanshuo Wang, Fufu Yu, Qiong Jia, and Shouhong Ding. Domain adaptive person
> search. In European conference on computer vision, pages 302–318. Springer, 2022.

---

### Official Review · Reviewer_WfPm · 2025-07-03

**Clarity:** 2
**Significance:** 2
**Originality:** 2
**Rating:** 4
**Confidence:** 3

**Summary:**

In this work, a person search framework, Spatiotemporal Consensus with Scene Prior (STCSP), is proposed to tackle the noise pseudo label issue in unsupervised learning. By leveraging spatio-temproal consensus procedure and scene prior-based method based on multiple previous approaches, the framework achieves the state-of-the-art performance on PRW and CUHK-SYSU.

**Questions:**

Compared to UDA baselines, the performance improvement of STCSP (ResNet-50) is more prominent on PRW than on CUHK-SYSU. Could you explain why?

**Ethical Concerns:**

["NO or VERY MINOR ethics concerns only"]

**Final Justification:**

After reviewing author's responses and other reviewers' comments, I think the author resolved my major concerns. Therefore, I decided to raise my rating score.

**Limitations:**

There is no limitation part in the paper.

**Paper Formatting Concerns:**

No concerns

**Quality:**

3

**Strengths And Weaknesses:**

The paper’s writing is coherent and it is easy to follow the work. The motivation for reducing noise on pseudo-labels is clear. Also, comprehensive experiments including an ablation study are properly conducted to showcase the proposed method.
However, the method appears to be a combination of several prior works, and the main contributions are not clearly distinguished. A more explicit explanation of what is novel and how it differs from existing approaches would be helpful. It might be because I'm not very familar with person search.

---

> ### Author Rebuttal · Authors · 2025-07-29
>
> _To Reviewer_ __WfPm__:
>
> __Q1:__ Compared to UDA baselines, the performance improvement of STCSP (ResNet-50) is more prominent on PRW than on CUHK-SYSU. Could you explain why?
>
> __A1:__ Thanks for this comment. The differential performance gains on PRW and CUHK-SYSU can be attributed to two key factors related to dataset characteristics and model design:
>
> - CUHK-SYSU is a larger dataset than PRW, with a greater number of images, identities, and scenes. The results on PRW are obtained through supervised-to-unsupervised transfer learning from CUHK-SYSU to PRW, which is a process of transferring from a large dataset to a small one. Our method achieves more significant performance improvements on PRW, demonstrating that it has a stronger knowledge transfer ability than other methods.
>
> - Since the images in PRW were taken on campus, some people wear the same clothing (such as activity T-shirts). Therefore, the pseudo-labels generated for PRW contain more noise than those for CUHK-SYSU, and our STCSP can effectively suppress this noise. This explains why the model's improvement on PRW is more pronounced.
>
> __Q2:__ More explicit explanation of the novelty.
>
> __A2:__ Thanks for this comment. Our method aims to effectively eliminate the interference of noise on pseudo-labels, establish positive feedback, and thus gradually bridge the domain gap for the UDA person search task. Different from previous works, we comprehensively consider the importance of both spatio and temporal consensus in the detection and re-identification stages, which establishes a complete consensus framework to suppress the noise originating from the domain gap and acquire knowledge from the target domain. Moreover, an iterative bilateral extremum matching method is designed. Within a few seconds, it can match the instances in any two of all thousands of images, thereby improving the performance of the model with small computational cost.

---

### Decision · Program_Chairs · 2025-09-17

**Decision:**

Accept (poster)

**Comment:**

This paper proposes a spatiotemporal consensus framework with a scene prior for unsupervised domain adaptive person search. Overall, the reviewers acknowledged the merits of the paper, including its novel integration of spatiotemporal consensus mechanisms and scene priors, as well as the state-of-the-art performance achieved on standard benchmarks. After rebuttal, all reviewers raised or maintained their positive ratings as the authors adequately addressed concerns regarding computational efficiency, methodological clarity, and experimental validation. The Area Chair recommends acceptance based on the technical novelty and strong empirical results, but encourages the authors to incorporate all reviewers' suggestions in the final version.